# Rising prevalence of parent-reported learning disabilities among U.S. children and adolescents aged 6–17 years: NSCH, 2016–2023

Chan Xu[1◉], Yanmei Li[2◉], Huijuan Yu[1], Qishan Li[1], Yingyu Liang[1], Kefan Zhou[1], Qian Li[1], Xinping Yu[3], Xia Zeng[1], Yabin Qu[4]*, Wenhan Yang◉[1]*

1 Department of Child and Adolescent Health, School of Public Health, Guangdong Pharmaceutical University, Guangzhou, Guangdong Province, China, 2 Nanshan District Center for Disease Control and Prevention (Nanshan District Health Supervision Institution), Shenzhen, Guangdong Province, China, 3 Zhongshan Ophthalmic Center, Sun Yat-sen University, Guangzhou, Guangdong Province, China, 4 Department of Environmental and School Health, Guangdong Provincial Center for Disease Control and Prevention, Guangzhou, Guangdong Province, China

◉ These authors contributed equally to this work.
* yabinqu@qq.com (YQ); wenhan-yang@gdpu.edu.cn (WY)

## Abstract

The prevalence of learning disabilities (LD) among children is a critical public health issue; however, recent LD prevalence estimates among children and adolescents aged 6–17 years, as reported by the National Survey of Children's Health (NSCH), remain largely unexplored. Data for this population-based cross-sectional study were obtained from NSCH to estimate the prevalence of LD diagnosis among U.S. children at both national and state levels, and to inspect the 8-year trends in these estimates from 2016 to 2023. Among 221,244 U.S. children, 20,644 had a history of LD diagnosis, with a weighted prevalence of 8.85% (95% $CI$ = 8.59–9.10). Of these, 19,289 were currently diagnosed with LD, yielding a weighted prevalence of 8.26% (95% $CI$ = 8.01–8.51). From 2016 to 2023, the prevalence of ever-diagnosed LD increased from 7.86% to 9.15%, while that of current-diagnosed LD rose from 7.32% to 8.66%, representing relative increases of 16.4% and 18.3%, respectively. The state with the highest prevalence (New Hampshire) had twice that of the state with the lowest prevalence (Utah). This study highlights a critical escalation in LD prevalence among U.S. children and adolescents between 2016 and 2023. Comprehensive screening and support programs must be implemented to enhance early identification and intervention.

## Introduction

Learning disability (LD), clinically termed Specific Learning Disorder (SLD) in modern diagnostic frameworks, refers to a heterogeneous group of neurodevelopmental conditions characterized by persistent difficulties in listening, speaking, reading, writing,

**Data availability statement:** All data files are available from the National Survey of Children's Health (NSCH) database (https://www.census.gov/programs-surveys/nsch.html).

**Funding:** The author(s) received no specific funding for this work.

**Competing interests:** The authors have declared that no competing interests exist.

reasoning, or mathematics abilities [1–3]. The conceptualization of LD aligns with the DSM-5 diagnostic criteria for Specific Learning Disorders [4] and ICD-11's Developmental Learning Disorders [5], which distinguish clinically significant learning disorders from general learning difficulties through requirements of: (a) persistent deficits lasting ≥ 6 months; (b) performance substantially below age expectations; (c) onset during developmental period; and (d) exclusion of alternative explanations (e.g., intellectual disability, inadequate instruction).

These disabilities are specific and life-long, potentially leading to long-term consequences in education, social interaction, and economic performance [6,7]. Compared with their peers without LD, children with LD often exhibit a higher vulnerability to mental and behavioral disorders—such as attention-deficit/hyperactivity disorder (ADHD), conduct disorders, anxiety, and depression, positing a continuous need for support and interventions throughout their lives [8,9].

Over the past decades, learning disabilities have emerged as a major health concern for school-aged children in the United States [10]. Literature reports indicated that the prevalence of LD among children varies from 8.2% to 9.7%, according to data extracted from the National Health Interview Survey (NHIS) between 1997–2021 and the National Survey of Children's Health in 2003 [10–12]. Both educators and clinicians have observed an increase in the number of children with learning disabilities. In a clinician-reported study, the prevalence of specific developmental delays, which include learning disabilities, speech, and language delays, surged by 40% from 1979 (1.5%) to 1996 (2.1%) among children aged between 4 and 15 years [13]. Nevertheless, a study indicated no significant annual average change in LD from the NHIS in the United States from 1997 to 2021 [12]. The reported prevalence of learning disabilities varies substantially across studies, often due to the barriers to identifying learning disabilities [14] and the heterogeneity of definitions, instruments used, and study designs [8,15]. Despite the reliability of NHIS studies, further scrutiny of more recent data is warranted to derive a deeper understanding of trends in learning disability prevalence.

The National Survey of Children's Health (NSCH) is a significant survey project that aims to investigate children's health and well-being. Since 2016, the NSCH has adopted redesigned surveys, moving from a periodic interviewer-assisted telephone survey to an annual self-administered web/paper-based survey utilizing an address-based sampling frame [16]. According to the topical questionnaire, children with special health care needs were more likely to be selected, bolstering the sample size of this targeted population [17]. Therefore, it is worth conducting health surveys to monitor the number and characteristics of children diagnosed with learning disabilities through NSCH data. This is crucial for the formulation of educational policies, early intervention and support, and the improvement of the learning environment for children.

In April 2024, The Maternal and Child Health Bureau in conjunction with the Census Bureau revised the imputation and weighting by race and ethnicity for the 2016–2021 NSCH. Following this revision, we have utilized the updated NSCH data to analyze nationally representative information in this study. Our objective is

to estimate the most recent prevalence of LD diagnosis among U.S. children and adolescents at both national and state levels and to inspect the 8-year trends in these estimates from 2016 to 2023.

## Methods

### Data source

Data for this study were obtained from the NSCH conducted between 2016 and 2023. The NSCH was a cross-sectional survey conducted via mail and online by the Data Resource Center (DRC) to collect health information on the welfare of children and adolescents aged 0–17 years in the United States [18]. The survey encompasses non-institutionalized children who live with their families or guardians and excludes those residing in institutional settings such as orphanages or group homes. Survey respondents were parents or caregivers with at least one child aged 0–17 years who resided in the same household during the entire interview. The funding and oversight for the NSCH were provided by the Health Resources and Services Administration's Maternal and Child Health Bureau [19]. Ethical approval for all survey procedures was obtained from the National Center for Health Statistics Research Ethics Review Board. Written consent was obtained through electronic submission or paper mailing, and it was returned to the U.S. Census Bureau [16]. The Guangdong Pharmaceutical University Academic Review Board determined the present study was exempt from approval because of the use of de-identified data. The NSCH data collection encompasses all 50 states and the District of Columbia, providing a comprehensive representation of the population in the United States [16].

### Data collection

The NSCH collected data on a variety of health topics through household interviews. For each family in the household, one child under 18 years of age was randomly selected as a sample. Information for this child was collected by interviewing an adult family member who was knowledgeable about the child's health. The NSCH employed complex weighting procedures to enhance national representativeness. The 2024 imputation revision by race/ethnicity addressed historical underrepresentation, improving demographic accuracy. Key quality indicators for 2016–2023 included: (a) the weighted topical completion rate of 27.1%–30.9% (reflecting full questionnaire completion among eligible households); (b) the weighted interview completion rate of 69.7%–78.5% (indicating cooperation among contacted households); and (c) the weighted overall response rate of 35.8%–39.1% (representing initial participation).

### Study population

Parents or caregivers were asked whether a doctor, other healthcare provider, or educator had ever told them that their child had a "Learning Disability" (Ever LD). Parents or caregivers who responded "Yes" were subsequently asked whether their child currently has "Learning Disability" (Current LD) [17]. It should be noted that "Current LD" represents parent-reported persistence of historically identified difficulties—not clinically confirmed active disorder. While consistent with surveillance methodologies [20], this approach may conflate ongoing impairment with historical diagnosis [21]. Because learning disabilities are not diagnosed until children enter school [4,22], analyses for this study were limited to 221,244 children aged 6–17 years and excluded children < 6 years ($n = 111,530$) [23]. Besides, 1,934 children were excluded because of missing data on ever-diagnosis information ($n = 803$), current-diagnosis information ($n = 256$), and highest parental education ($n = 875$).

### Study variables

The outcome variables for this analysis were "Ever LD" and "Current LD." Information on sex, race/ethnicity, family income, and highest education of family members were collected using a standardized questionnaire during the interview. The study divided the race/ethnicity into four groups: Hispanic, non-Hispanic white, non-Hispanic black,

and other based on the 1997 Office of Management and Budget Standards [24]. The highest educational level of family members was categorized as less than high school, high school, and college or higher. Family income factors included in the analysis were federal poverty level (FPL), grouped as 0–99% FPL, 100–199% FPL, 200–399% FPL, and ≥ 400% FPL [25].

### Statistical analysis

Weighted estimates were applied according to the NSCH Analytic Guidelines [17]. Population characteristics were shown according to ever and current LD prevalence with 95% confidence interval (CI). Rao–Scott chi-square tests were employed to compare differences of categorical variables. A logistic regression model was used to estimate trends in the prevalence over time, which included the survey year as a continuous variable and adjusted for age, sex, race/ethnicity, family income, and highest education of family members. All of the statistical analyses were performed using SAS Software, version 9.4. All estimates are weighted based on the complex sampling design of NSCH. The 2-sided statistical significance level was set at α = 0.05.

## Results

### Prevalence of parent-reported LD in U.S. children and adolescents, 2016–2023

This study included 221,244 children and adolescents aged 6–17 years (114,447 boys [51.11%] and 106,797 girls [48.89%]; 29,287 Hispanic [26.45%], 148,653 non-Hispanic white [49.36%], 14,683 non-Hispanic black [13.63%], and 28,621 other race/ethnicity [10.56%] individuals) for the period 2016–2023. Of the eligible children, 20,644 were reported to have ever been diagnosed with an LD and the weighted prevalence was 8.85% (95% *CI*, 8.59–9.10). Among those who had ever been diagnosed as having an LD, 19,289 were reported to have currently been diagnosed with an LD and the weighted prevalence was 8.26% (95% *CI*, 8.01–8.51) (Table 1). The state with the highest prevalence, New Hampshire (12.84% of ever-diagnosed LD and 11.86% of current LD), had twice that of the state with the lowest prevalence, Utah (6.18% of ever-diagnosed LD and 5.82% of current LD). See Figs 1 and 2.

We found that age, sex, race/ethnicity, family income, and highest education of family members were significant differences in both Ever LD and Current LD groups.

The weighted prevalence of LD was slightly higher in children aged 12–17 years (Ever LD, 9.74%, 95% *CI*, 9.39–10.10; Current LD, 8.96%, 95% *CI*, 8.62–9.29) than in children aged 6–11 years (Ever LD, 7.92%, 95% *CI*, 7.55–8.30; Current LD, 7.54%, 95% *CI*, 7.18–7.91). With regard to children's sex, the weighted prevalence of LD was higher in boys (Ever LD, 10.75%, 95% *CI*, 10.36–11.14; Current LD, 9.99%, 95% *CI*, 9.62–10.36) than in girls (Ever LD, 6.85%, 95% *CI*, 6.52–7.19; Current LD, 6.45%, 95% *CI*, 6.13–6.78). For race/ethnicity and level of family education, we found that the weighted prevalence of LD was highest among non-Hispanic blacks (Ever LD, 11.47%, 95% *CI*, 10.59–12.36; Current LD, 10.64%, 95% *CI*, 9.79–11.49) and those with a high school education (Ever LD, 11.48%, 95% *CI*, 10.75–12.21; Current LD, 10.79%, 95% *CI*, 10.08–11.50). The weighted ever and current LD prevalence increase with the decrease of family economic level (Table 1).

### Trends in ever and current LD in U.S. children and adolescents, 2016–2023

Significant increasing trends in learning disability prevalence were observed from 2016 to 2023. The weighted prevalence of Ever LD rose from 7.86% (95% *CI*, 7.25–8.46) to 9.15% (95% *CI*, 8.55–9.75; *p* for trend < 0.001), while Current LD increased from 7.32% (95% *CI*, 6.73–7.91) to 8.66% (95% *CI*, 8.06–9.25; *p* for trend < 0.001), representing relative increases of 16.4% and 18.3% respectively. The steepest increases in both Ever LD and Current LD prevalence occurred among adolescents aged 12–17 years, females, non-Hispanic Whites, and children from households with college-educated parents or incomes ≥ 200% FPL. See Tables 2 and 3 and Fig 3.

**Table 1. Prevalence of diagnosed LD in U.S. children and adolescents aged 6–17 years, 2016–2023.**

| Characteristic | Overall Participants, No.(%)a | Ever LD | | | Current LD | | |
|---|---|---|---|---|---|---|---|
| | | Participants, No. (%)a | Prevalence, % (95% CI)b | p Valuec | Participants, No. (%)a | Prevalence, % (95% CI)b | p Valuec |
| Overall | 221,244 | 20,644 | 8.85 (8.59–9.10) | | 19,289 | 8.26 (8.01–8.51) | |
| Age, year | | | | | | | |
| 6–11 | 96,807 (49.29) | 7,924 (38.38) | 7.92 (7.55–8.30) | < 0.001 | 7,574 (39.27) | 7.54 (7.18–7.91) | < 0.001 |
| 12–17 | 124,437 (50.71) | 12,720 (61.62) | 9.74 (9.39–10.10) | | 11,715 (60.73) | 8.96 (8.62–9.29) | |
| Sex | | | | | | | |
| Male | 114,447 (51.11) | 12,941 (62.69) | 10.75 (10.36–11.14) | < 0.001 | 12,065 (62.55) | 9.99 (9.62–10.36) | < 0.001 |
| Female | 106,797 (48.89) | 7,703 (37.31) | 6.85 (6.52–7.19) | | 7,224 (37.45) | 6.45 (6.13–6.78) | |
| Race/ethnicityd | | | | | | | |
| Hispanic | 29,287 (26.45) | 2,749 (13.32) | 8.25 (7.58–8.91) | < 0.001 | 2,548 (13.21) | 7.63 (6.98–8.27) | < 0.001 |
| Non-Hispanic white | 148,653 (49.36) | 13,898 (67.32) | 8.79 (8.52–9.06) | | 13,003 (67.41) | 8.25 (7.99–8.51) | |
| Non-Hispanic black | 14,683 (13.63) | 1,751 (8.48) | 11.47 (10.59–12.36) | | 1,633 (8.47) | 10.64 (9.79–11.49) | |
| Other | 28,621 (10.56) | 2,246 (10.88) | 7.23 (6.65–7.81) | | 2,105 (10.91) | 6.84 (6.28–7.41) | |
| Highest educational level of family members | | | | | | | |
| Less than high school | 6,125 (10.22) | 635 (3.08) | 8.25 (7.07–9.43) | < 0.001 | 596 (3.09) | 7.68 (6.55–8.82) | < 0.001 |
| High school | 30,210 (19.79) | 3,772 (18.27) | 11.48 (10.75–12.21) | | 3,539 (18.35) | 10.79 (10.08–11.50) | |
| College or higher | 184,909 (69.99) | 16,237 (78.65) | 8.19 (7.94–8.44) | | 15,154 (78.56) | 7.63 (7.39–7.87) | |
| Family income, % FPLe | | | | | | | |
| 0–99 | 26,654 (19.09) | 3,717 (18.01) | 12.16 (11.34–12.98) | < 0.001 | 3,513 (18.21) | 11.38 (10.59–12.17) | < 0.001 |
| 100–199 | 36,047 (21.22) | 4,165 (20.18) | 10.25 (9.60–10.90) | | 3,889 (20.16) | 9.55 (8.91–10.18) | |
| 200–399 | 66,671 (28.36) | 6,078 (29.44) | 8.07 (7.65–8.49) | | 5,663 (29.36) | 7.50 (7.10–7.90) | |
| ≥400 | 91,872 (31.33) | 6,684 (32.38) | 6.58 (6.28–6.88) | | 6,224 (32.27) | 6.17 (5.88–6.46) | |

Abbreviation: LD = learning disability; FPL = federal poverty level; CI = confidence interval.

aThe numbers of participants were unweighted.

bPrevalence estimates were weighted.

cP value for overall differences in prevalence by stratum.

dRace and Hispanic ethnicity were self-reported and classified based on 1997 U.S. Office of Management and Budget standards. Other races and ethnicities included non-Hispanic American Indian or Alaska Native, non-Hispanic Asian, non-Hispanic Native Hawaiian and Other Pacific Islander, and non-Hispanic multiple race.

eThe family income-to-poverty ratio is the total family income divided by the poverty threshold.

## Discussion

In a nationally representative population from the NSCH, we identified a weighted prevalence of ever-diagnosed LD at 8.85% and current-diagnosed LD at 8.26% among U.S. children who were aged 6–17 years in the period 2016–2023. From 2016 to 2023, there was a significant increase in the prevalence of both ever and current LD over the past 8 years, with ever LD prevalence rising significantly from 7.86% to 9.15% (a 16.4% increase) and current LD prevalence rising significantly from 7.32% to 8.66% (an 18.3% increase). In the subgroups, we also found a significant increase in the average annual change in the prevalence of both ever LD and current LD among those aged 12–17 years, females, non-Hispanic whites, individuals with a college education or higher, and those with a family income of 200% FPL or above.

The overall prevalence of LD in this study was slightly higher than those reported in other studies. A study using NSCH data from 2018–2020 to examine the relationship between screen time and developmental disabilities among children aged 0–17 years reported a 7.2% prevalence of LD [26]. Similarly, another study conducted by Zablotsky, which utilized

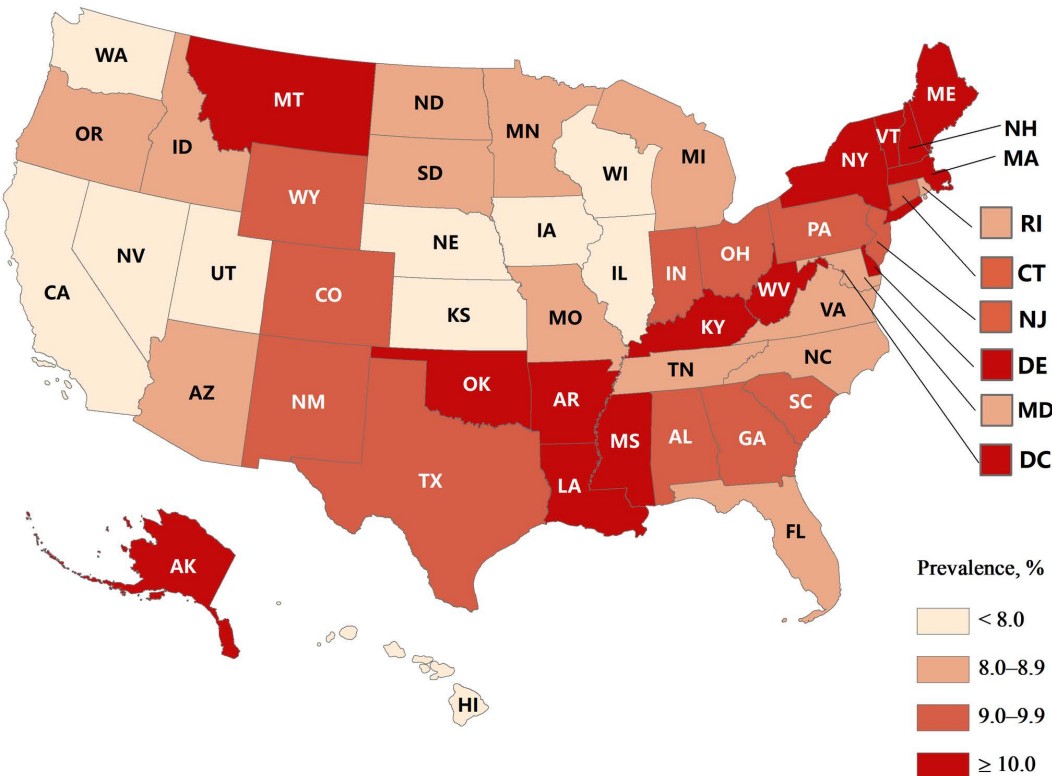

**Fig 1. Weighted prevalence estimates of parent-reported ever learning disability (LD) among children aged 6–17 years by state (United States, 2016–2023).** Base map administrative boundaries sourced from the U.S. Census Bureau TIGER/Line Shapefiles (public domain, available at: [https://catalog.data.gov/dataset/tiger-line-shapefile-current-nation-u-s-state-and-equivalent-entities]). Map created by authors.

NHIS data from 2009–2017, estimated the prevalence at 7.74% among U.S. children aged 3–17 years [27]. Other research using NHIS data between 1997–2021 found lower LD prevalence estimates, ranging from 7.62% to 8.31%, compared to our current study [12,28]. Apart from the overall prevalence, we also found the prevalence differed significantly by age, sex, race/ethnicity, family education, and family income levels from 2016 to 2023, which was similar to other findings [10,27]. State-based estimates ranged from 6.18% in Utah to 12.84% in New Hampshire.

Learning disabilities are difficult to diagnose, because the age at identification depends not only on the type and severity of the learning disabilities, but also on whether there are associated deficits [29]. Moreover, learning disabilities are often considered lifelong conditions [8]. Older children may exhibit a higher and increasing prevalence rate, likely due to their longer exposure to evaluation and the possibility of being diagnosed [10]. Beyond age-related patterns, several key demographic factors have been consistently linked to disparities in LD prevalence and diagnosis, including gender, racial/ethnic background, and family socioeconomic status (SES). This section discusses these factors in turn.

Nearly all studies have reported a higher prevalence of LD and other developmental disorders in boys than in girls [30]. Studies on gender discrepancies in LD have noted that females tend to have advantages in verbal working memory and literacy, with better linguistic processing and reading fluency—factors that may contribute to underdiagnosis in girls relative to boys [31–33].

Additionally, our findings indicated that non-Hispanic blacks and non-Hispanic whites are more likely to be diagnosed with learning disabilities, with an increasing trend observed among non-Hispanic whites. This aligns with findings from

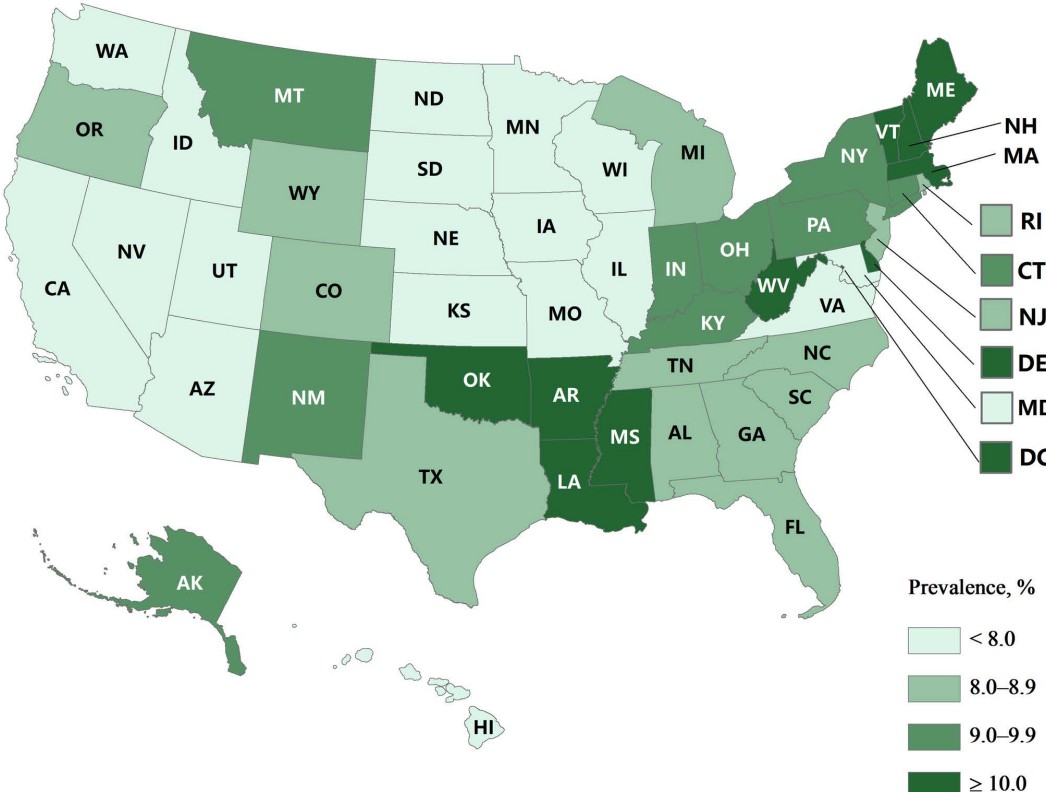

**Fig 2. Weighted prevalence estimates of parent-reported current learning disability (LD) among children aged 6–17 years by state (United States, 2016–2023).** Base map administrative boundaries sourced from the U.S. Census Bureau TIGER/Line Shapefiles (public domain, available at: [https://catalog.data.gov/dataset/tiger-line-shapefile-current-nation-u-s-state-and-equivalent-entities]). Map created by authors.

other studies [34,35]. Previous literature has suggested that limited access to healthcare services and language barriers may account for the lower reported incidence of LD among Hispanic parents [36].

Consistent with other reports, LD is associated with family socioeconomic status. Lower household education and lower family income levels are linked to higher odds of LD [10,11,37]. Nicole et al. further suggested a connection between low socioeconomic status and diminished cognitive, academic, and behavioral performance in early childhood [38,39]. Notably, an increasing trend of LD was observed in families with higher education levels (e.g., college education or beyond). Although the reasons for this trend remain unclear, it is possible that higher parental expectations and demands may lead to over-reporting of LD in this group.

Over the 8-year period from 2016 to 2023, our study revealed a significant annual average change in the prevalence of both ever-diagnosed and currently-diagnosed LD. From 2016 to 2023, the prevalence of ever-diagnosed LD increased from 7.86% to 9.15%, while that of current-diagnosed LD rose from 7.32% to 8.66%, representing relative increases of 16.4% and 18.3%, respectively. However, such a significant rise was not observed in other studies pertaining to LD [12,27]. Over the past twenty-five years, the prevalence of LD, according to NHIS data, was recorded at 7.66% from 1997 to 2008 [40], 7.74% from 2009 to 2017 [27], and 7.6% from 2019 to 2020 [41], respectively. The discrepancy observed between learning disabilities estimates derived from NSCH data and those from NHIS data may partially stem from differences in data collection methods and the sample size between the two databases [42].

Non-etiological factors may be partly responsible for the increase in the prevalence of LD in this study. We estimated the prevalence of parent-reported LD using data from the NSCH; however, since the questionnaire only inquired about

**Table 2. Trends in the prevalence of ever diagnosed LD in U.S. children and adolescents aged 6 to 17 years, 2016–2023.**

| Characteristic | Prevalence of Ever LD, % (95% CI)[a] | | | | | | | | p for trend[b] |
|---|---|---|---|---|---|---|---|---|---|
| | 2016 | 2017 | 2018 | 2019 | 2020 | 2021 | 2022 | 2023 | |
| No. of participants overall[c] | 34,686 | 15,378 | 21,848 | 21,168 | 30,495 | 30,046 | 34,184 | 33,439 | NA |
| No. of participants with ever LD[c] | 2,909 | 1,361 | 1,935 | 2,860 | 2,860 | 2,970 | 3,410 | 3,265 | NA |
| Overall | 7.86 | 9.08 | 8.60 | 8.11 | 9.22 | 9.25 | 9.45 | 9.15 | < 0.001 |
| | (7.25–8.46) | (7.99–10.16) | (7.84–9.36) | (7.42–8.80) | (8.53–9.92) | (8.58–9.92) | (8.86–10.05) | (8.55–9.75) | |
| **Age, year** | | | | | | | | | |
| 6–11 | 6.94 | 8.97 | 8.36 | 7.02 | 7.41 | 8.05 | 8.41 | 8.20 | 0.150 |
| | (6.10–7.77) | (7.23–10.70) | (7.23–9.49) | (6.09–7.94) | (6.56–8.27) | (7.17–8.93) | (7.54–9.28) | (7.33–9.07) | |
| 12–17 | 8.79 | 9.18 | 8.84 | 9.19 | 10.96 | 10.41 | 10.43 | 10.04 | < 0.001 |
| | (7.91–9.66) | (7.89–10.48) | (7.83–9.85) | (8.17–10.20) | (9.89–12.04) | (9.40–11.42) | (9.60–11.26) | (9.21–10.87) | |
| **Sex** | | | | | | | | | |
| Male | 9.55 | 11.28 | 10.57 | 10.14 | 11.21 | 11.50 | 10.74 | 10.99 | 0.035 |
| | (8.72–10.38) | (9.68–12.89) | (9.40–11.74) | (9.09–11.19) | (10.13–12.29) | (10.41–12.59) | (9.92–11.57) | (10.09–11.89) | |
| Female | 6.10 | 6.77 | 6.55 | 6.00 | 7.15 | 6.90 | 8.10 | 7.22 | 0.003 |
| | (5.21–6.99) | (5.32–8.21) | (5.60–7.49) | (5.12–6.88) | (6.30–8.01) | (6.15–7.65) | (7.23–8.98) | (6.44–8.01) | |
| **Race/ethnicity** | | | | | | | | | |
| Hispanic | 6.88 | 10.02 | 8.58 | 7.22 | 8.12 | 8.36 | 8.35 | 8.37 | 0.927 |
| | (5.27–8.48) | (6.95–13.09) | (6.61–10.56) | (5.54–8.91) | (6.41–9.82) | (6.56–10.16) | (6.95–9.75) | (7.08–9.65) | |
| Non-Hispanic white | 8.14 | 7.99 | 8.27 | 8.33 | 9.20 | 9.51 | 9.67 | 9.26 | < 0.001 |
| | (7.48–8.80) | (7.12–8.86) | (7.51–9.04) | (7.53–9.12) | (8.43–9.97) | (8.80–10.22) | (8.95–10.39) | (8.56–9.97) | |
| Non-Hispanic black | 10.34 | 12.19 | 10.83 | 9.95 | 12.68 | 11.30 | 12.27 | 12.20 | 0.168 |
| | (8.42–12.25) | (8.59–15.80) | (8.28–13.38) | (7.69–12.22) | (10.35–15.01) | (9.26–13.34) | (10.15–14.39) | (9.76–14.63) | |
| Other | 5.47 | 7.54 | 7.16 | 6.90 | 7.64 | 7.58 | 8.05 | 7.15 | 0.102 |
| | (4.30–6.64) | (4.67–10.40) | (5.44–8.88) | (5.36–8.43) | (5.97–9.30) | (6.01–9.15) | (6.81–9.28) | (6.06–8.25) | |
| **Highest educational level of family members** | | | | | | | | | |
| Less than high school | 8.45 | 12.35 | 7.09 | 7.05 | 9.70 | 5.71 | 8.47 | 6.95 | 0.215 |
| | (5.60–11.30) | (6.53–18.17) | (4.36–9.82) | (4.56–9.54) | (6.56–12.83) | (3.11–8.32) | (6.10–10.84) | (4.42–9.48) | |
| High school | 10.14 | 12.61 | 11.80 | 10.83 | 11.18 | 11.90 | 11.62 | 11.75 | 0.398 |
| | (8.43–11.85) | (9.59–15.63) | (9.68–13.91) | (8.79–12.78) | (9.26–13.10) | (9.91–13.90) | (9.87–13.37) | (10.22–13.29) | |
| College or higher | 7.07 | 7.55 | 7.96 | 7.51 | 8.58 | 9.00 | 9.01 | 8.73 | < 0.001 |
| | (6.51–7.63) | (6.63–8.47) | (7.15–8.76) | (6.81–8.22) | (7.90–9.27) | (8.33–9.68) | (8.39–9.63) | (8.07–9.38) | |
| **Family income, % FPL** | | | | | | | | | |
| 0–99 | 10.93 | 14.35 | 10.90 | 12.24 | 12.16 | 12.24 | 11.97 | 12.46 | 0.777 |
| | (9.07–12.80) | (10.68–18.02) | (8.92–12.88) | (9.85–14.62) | (10.18–14.14) | (10.25–14.22) | (10.43–13.50) | (10.39–14.52) | |
| 100–199 | 8.88 | 11.18 | 10.46 | 9.24 | 10.54 | 10.89 | 10.73 | 10.16 | 0.479 |
| | (7.33–10.44) | (8.44–13.91) | (8.47–12.46) | (7.64–10.84) | (8.92–12.17) | (9.07–12.70) | (9.15–12.31) | (8.87–11.45) | |
| 200–399 | 6.87 | 7.15 | 7.92 | 7.02 | 9.18 | 7.85 | 9.44 | 8.86 | < 0.001 |
| | (5.99–7.76) | (5.82–8.47) | (6.47–9.38) | (6.01–8.02) | (7.79–10.58) | (6.87–8.83) | (8.23–10.66) | (7.83–9.90) | |
| ≥ 400 | 5.81 | 5.79 | 6.40 | 5.93 | 6.60 | 7.66 | 7.32 | 6.98 | < 0.001 |
| | (5.15–6.48) | (4.77–6.82) | (5.52–7.29) | (5.04–6.83) | (5.79–7.40) | (6.72–8.60) | (6.57–8.06) | (6.22–7.73) | |

[a]Prevalence estimates were weighted.

[b]P value for trends were calculated using weighted logistic regression models, which included survey year as a continuous variable and adjusted for age, sex, race/ethnicity, education, and family income to poverty ratio.

[c]The numbers of participants overall and with LD were unweighted.

**Table 3. Trends in the prevalence of current diagnosed LD in U.S. children and adolescents aged 6 to 17 years, 2016–2023.**

| Characteristic | Prevalence of Current LD, % (95% *CI*)[a] | | | | | | | | p for trend[b] |
|---|---|---|---|---|---|---|---|---|---|
| | **2016** | **2017** | **2018** | **2019** | **2020** | **2021** | **2022** | **2023** | |
| No. of participants overall[c] | 34,686 | 15,378 | 21,848 | 21,168 | 30,495 | 30,046 | 34,184 | 30,174 | NA |
| No. of participants with current LD[c] | 2,704 | 1,275 | 1,792 | 1,794 | 2,694 | 2,776 | 3,186 | 3,068 | NA |
| Overall | 7.32 | 8.69 | 7.98 | 7.37 | 8.71 | 8.53 | 8.80 | 8.66 | < 0.001 |
| | (6.73–7.91) | (7.61–9.76) | (7.24–8.71) | (6.74–8.00) | (8.02–9.39) | (7.90–9.15) | (8.22–9.38) | (8.06–9.25) | |
| Age, year | | | | | | | | | |
| 6–11 | 6.54 | 8.79 | 7.82 | 6.62 | 7.00 | 7.57 | 8.08 | 7.88 | 0.163 |
| | (5.74–7.34) | (7.06–10.53) | (6.72–8.92) | (5.73–7.50) | (6.16–7.84) | (6.72–8.42) | (7.22–8.94) | (7.02–8.75) | |
| 12–17 | 8.10 | 8.58 | 8.13 | 8.11 | 10.35 | 9.44 | 9.48 | 9.38 | < 0.001 |
| | (7.24–8.96) | (7.31–9.85) | (7.16–9.09) | (7.21–9.00) | (9.29–11.41) | (8.54–10.35) | (8.70–10.26) | (8.56–10.19) | |
| Sex | | | | | | | | | |
| Male | 8.94 | 10.81 | 9.73 | 9.12 | 10.61 | 10.44 | 9.83 | 10.39 | 0.071 |
| | (8.14–9.73) | (9.22–12.40) | (8.61–10.85) | (8.16–10.09) | (9.54–11.67) | (9.46–11.43) | (9.05–10.60) | (9.50–11.27) | |
| Female | 5.63 | 6.46 | 6.14 | 5.54 | 6.72 | 6.52 | 7.73 | 6.84 | < 0.001 |
| | (4.76–6.51) | (5.03–7.90) | (5.22–7.06) | (4.74–6.34) | (5.89–7.55) | (5.79–7.26) | (6.86–8.59) | (6.06–7.62) | |
| Race/ethnicity | | | | | | | | | |
| Hispanic | 6.48 | 9.72 | 7.74 | 6.51 | 7.61 | 7.24 | 7.67 | 7.97 | 0.924 |
| | (4.90–8.05) | (6.66–12.79) | (5.83–9.65) | (4.98–8.04) | (5.93–9.30) | (5.68–8.80) | (6.35–9.00) | (6.70–9.24) | |
| Non-Hispanic white | 7.64 | 7.55 | 7.76 | 7.65 | 8.66 | 8.89 | 9.18 | 8.69 | < 0.001 |
| | (6.99–8.28) | (6.70–8.40) | (7.01–8.50) | (6.90–8.40) | (7.91–9.41) | (8.20–9.58) | (8.47–9.89) | (8.00–9.38) | |
| Non-Hispanic black | 9.35 | 11.56 | 9.99 | 8.54 | 12.15 | 10.73 | 11.10 | 11.62 | 0.131 |
| | (7.62–11.09) | (7.99–15.12) | (7.60–12.39) | (6.56–10.52) | (9.84–14.45) | (8.74–12.72) | (9.07–13.13) | (9.20–14.04) | |
| Other | 4.98 | 7.48 | 6.86 | 6.65 | 7.18 | 7.16 | 7.39 | 6.79 | 0.160 |
| | (3.84–6.12) | (4.61–10.35) | (5.16–8.57) | (5.12–8.17) | (5.61–8.75) | (5.61–8.70) | (6.21–8.57) | (5.72–7.86) | |
| Highest educational level of family members | | | | | | | | | |
| Less than high school | 7.83 | 11.98 | 6.50 | 6.57 | 9.38 | 4.48 | 7.99 | 6.50 | 0.174 |
| | (5.12–10.54) | (6.18–17.78) | (3.89–9.10) | (4.17–8.97) | (6.27–12.49) | (2.94–6.02) | (5.66–10.33) | (4.00–9.00) | |
| High school | 9.68 | 12.04 | 11.08 | 9.67 | 10.81 | 11.03 | 10.91 | 11.12 | 0.473 |
| | (7.99–11.36) | (9.05–15.03) | (9.00–13.15) | (7.78–11.56) | (8.90–12.73) | (9.11–12.95) | (9.24–12.59) | (9.61–12.63) | |
| College or higher | 6.52 | 7.21 | 7.35 | 6.85 | 7.99 | 8.39 | 8.35 | 8.26 | < 0.001 |
| | (5.98–7.06) | (6.30–8.12) | (6.58–8.12) | (6.21–7.48) | (7.33–8.65) | (7.74–9.04) | (7.75–8.95) | (7.62–8.91) | |
| Family income, % FPL | | | | | | | | | |
| 0–99 | 10.06 | 13.75 | 10.18 | 11.32 | 11.53 | 10.94 | 11.26 | 12.01 | 0.712 |
| | (8.29–11.82) | (10.10–17.40) | (8.26–12.11) | (9.11–13.53) | (9.58–13.48) | (9.30–12.58) | (9.76–12.76) | (9.96–14.07) | |
| 100–199 | 8.15 | 10.78 | 9.76 | 8.41 | 10.00 | 10.15 | 9.61 | 9.54 | 0.627 |
| | (6.63–9.68) | (8.07–13.49) | (7.82–11.69) | (6.92–9.90) | (8.39–11.60) | (8.38–11.91) | (8.16–11.06) | (8.28–10.80) | |
| 200–399 | 6.59 | 6.72 | 7.12 | 6.15 | 8.73 | 7.22 | 8.88 | 8.37 | < 0.001 |
| | (5.71–7.47) | (5.42–8.02) | (5.75–8.48) | (5.31–6.99) | (7.35–10.11) | (6.29–8.15) | (7.68–10.09) | (7.35–9.40) | |
| ≥400 | 5.43 | 5.58 | 6.04 | 5.48 | 6.10 | 7.19 | 6.89 | 6.52 | < 0.001 |
| | (4.78–6.07) | (4.57–6.60) | (5.17–6.91) | (4.61–6.34) | (5.34–6.86) | (6.27–8.12) | (6.17–7.61) | (6.79–7.26) | |

[a]Prevalence estimates were weighted.

[b]P value for trends were calculated using weighted logistic regression models, which included survey year as a continuous variable and adjusted for age, sex, race/ethnicity, education, and family income to poverty ratio.

[c]The numbers of participants overall and with LD were unweighted.

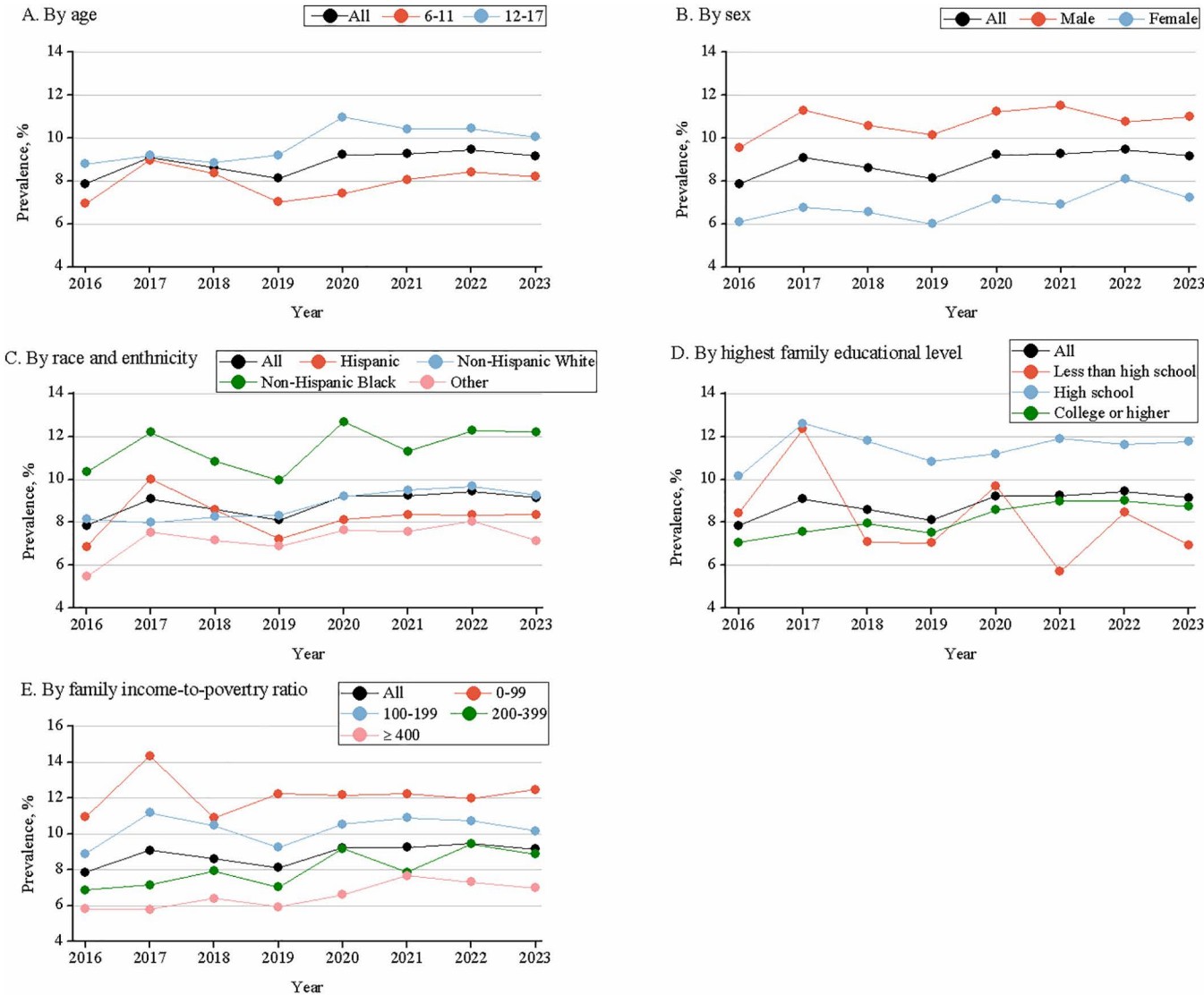

**Fig 3. Trends in the prevalence of ever learning disability (LD), 2016–2023.** Prevalence estimates were weighted and are among children and adolescents aged 6–17 years. The weighted prevalence of Ever LD was 7.86% (95% *CI*, 7.25–8.46) in 2016, 9.08% (95% *CI*, 7.99–10.16) in 2017, 8.60% (95% *CI*, 7.84–9.36) in 2018, 8.11% (95% *CI*, 7.42–8.80) in 2019, 9.22% (95% *CI*, 8.53–9.92) in 2020, 9.25% (95% *CI*, 8.58–9.92) in 2021, 9.45% (95% *CI*, 8.86–10.05) in 2022, and 9.15% (95% *CI*, 8.55–9.75) in 2023, respectively. For trend, *p* was calculated using a weighted logistic regression model, which included survey year as a continuous variable and was adjusted for age (6–11 years: *p* = 0.150; 12–17 years: *p* < 0.001), sex (male: *p* = 0.035; female: *p* = 0.003), race and ethnicity (Hispanic: *p* = 0.927; non-Hispanic White: *p* < 0.001; non-Hispanic Black: *p* = 0.168; other: *p* = 0.102), highest educational level of family members (less than high school: *p* = 0.215; high school: *p* = 0.398; college or higher: *p* < 0.001), and family income-to-poverty ratio (< 1.00: *p* = 0.777; 1.00–1.99: *p* = 0.479; 2.00–3.99: *p* < 0.001; ≥ 4.00: *p* < 0.001).

"learning disabilities" without specifying which disorders were regarded as LD, parents could have comprehended this term to describe an extensive range of learning difficulties [10,43]. The expansion of medical education and a growing emphasis on learning disabilities has led to an increased sensitivity of doctors towards diagnosing these conditions, as well as changes in the acceptance of the diagnosis of learning disabilities by parents and teachers, which have led to an increase in the prevalence of learning disabilities [13,14]. The availability of resources and the presence of other

developmental disabilities can also significantly impact the number and prevalence of LD [44]. As the prevalence of disorders such as autism spectrum disorder (ASD) and attention-deficit hyperactivity disorder (ADHD) increases, so does the prevalence of LD due to the comorbidity of LD and ASD/ADHD [10,45].

Contemporary research has established multifactorial etiologies for learning disabilities, integrating genetic vulnerabilities, neurocognitive mechanisms, and environmental factors [46,47]. Certain highly penetrant heritable disorders(where > 95% of variant carriers develop symptoms) still show environmentally modulated clinical expression [48]. In terms of environmental triggers, prenatal and perinatal risk factors, including prematurity, low birthweight, maternal smoking, and maternal alcohol abuse during pregnancy, have been associated with LD risk [10,49–51]. Stein et al. conducted an analysis of NHIS data involving 7,817 children and discovered that those with moderately low birthweight were significantly more likely to be diagnosed with a learning disability compared to their normal birthweight counterparts [52]. Many substantiated or implicated environmental-biological factors are associated with learning disabilities, such as cadmium, phthalates, and lead [53,54]. A study using data from the National Health and Nutrition Examination Survey (NHANES) showed that children in the highest quartile of urinary cadmium concentration had significantly higher odds of having a self-reported LD when compared with those in the lowest quartile [53,55].

Children with learning disabilities may experience social exclusion, bullying, poor self-image, and underachievement [56]. Research indicates that many learning disabilities can have long-term, often lifelong impacts, presenting numerous challenges across various functional areas that adults with these conditions must navigate—including employment, family dynamics, social and emotional well-being, daily living routines, community participation, and recreational activities [11,57]. Meanwhile, it is noteworthy that a family's level of education bears an impact on their economic status. Children living in poverty tend to score lower on standardized tests of academic achievement, a trend that continues into adulthood, subsequently leading to lower wages and incomes [39]. In addition, the economic impact of increased LD prevalence on families, schools, and the healthcare system is significant [58–60]. Using data from the National Audit Commission and the Department of Health, Forsyth et al. reported a widespread discrepancy from the median spend/ burden ratio standing at £10,260 per person with a learning disability [61]. According to a study conducted in Mumbai, the direct, indirect, and intangible costs associated with a specific learning disability amounted to Indian Rupees (INR) 5,936,053, 29,261,220, and 42,295,000, respectively [62]. In the United States, over half of students receiving special education services receive them due to learning disabilities—a statistic that underscores the substantial resource allocation directed toward supporting these individuals within educational systems [10]. Given the significant economic and social costs associated with LD, it is crucially important to continuously monitor the prevalence and characteristics of children diagnosed with these conditions.

## Strengths and limitations

The main advantage of this study is the use of a nationally representative sample of the U.S. population, which allows us to generalize our findings to a wider population. Furthermore, by employing a series of nationwide, population-based, cross-sectional surveys, not only were we able to examine the secular trends in the prevalence of learning disabilities over a period of 8 years, but it also facilitated an assessment of the disparities in this prevalence relative to various population characteristics.

This study has several limitations. First, the cross-sectional design of the NSCH precludes establishing causal relationships between specific factors and LD diagnosis, limiting our ability to infer directional influences. Second, the reliance on parental reports for the diagnosis of LD may introduce inaccuracies due to misreporting and recall bias. Parents may interpret "learning disability" broadly—encompassing sensory/motor issues, intellectual disability, or environmental factors—rather than specific subtypes (e.g., reading/writing/math difficulties). This potential overbreadth is suggested by the relatively high co-occurrence of parent-reported LD with intellectual disability or other developmental delays in our sample, which may reflect overlapping interpretations of the term. Additionally, without verifying parental reports against school or

health records, the accuracy of these diagnoses remains unconfirmed. Third, our operationalization of LD is undifferentiated, encompassing a broad range of subtypes without distinguishing between them; this may obscure subtype-specific patterns and limit the granularity of our findings. Finally, the U.S.-focused sample raises generalizability concerns, particularly given the WEIRD (Western, Educated, Industrialized, Rich, Democratic) bias [63], as global differences in cultural, educational, and socioeconomic contexts may shape LD identification in ways not reflected here. These limitations necessitate cautious interpretation of the findings.

## Conclusions

This study presents a national and state-level overview of the epidemiology and characteristics of school-aged children with parent-reported diagnoses of LD from 2016 to 2023, and reveals a 16.4–18.3% increase in parent-reported LD prevalence among U.S. children and adolescents over this period. The total weighted prevalence was 8.85% for ever LD and 8.26% for current LD, with notable disparities across age, sex, and states. Given the shifting trends in LD prevalence and its far-reaching impacts, continuous monitoring of epidemiological patterns remains critical. Furthermore, cross-cultural research comparing LD identification, support systems, and societal perceptions—bridging insights from individualistic contexts such as the United States and collectivist traditions such as China—could enhance global understanding of how cultural values shape learning disability outcomes.

## Author contributions

**Conceptualization:** Yanmei Li, Xinping Yu, Xia Zeng, Yabin Qu, Wenhan Yang.

**Data curation:** Chan Xu, Yanmei Li, Huijuan Yu, Qishan Li, Yingyu Liang, Kefan Zhou, Qian Li.

**Formal analysis:** Chan Xu.

**Methodology:** Chan Xu, Yanmei Li, Huijuan Yu, Qishan Li, Yingyu Liang, Kefan Zhou.

**Resources:** Qian Li, Wenhan Yang.

**Software:** Chan Xu, Yanmei Li, Huijuan Yu, Qishan Li, Yingyu Liang, Kefan Zhou, Qian Li, Wenhan Yang.

**Supervision:** Xinping Yu, Xia Zeng, Yabin Qu, Wenhan Yang.

**Validation:** Yabin Qu, Wenhan Yang.

**Visualization:** Xinping Yu, Yabin Qu, Wenhan Yang.

**Writing – original draft:** Chan Xu, Yanmei Li.

**Writing – review & editing:** Chan Xu, Xia Zeng, Yabin Qu, Wenhan Yang.

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
