## [Decision Letter · Decision Letter 0]

1 Jun 2025

Dear Dr. Yang,

Thank you for submitting your manuscript to PLOS ONE. After careful consideration, we feel that it has merit but does not fully meet PLOS ONE’s publication criteria as it currently stands. Therefore, we invite you to submit a revised version of the manuscript that addresses the points raised during the review process.

Please submit your revised manuscript by Jul 16 2025 11:59PM. If you will need more time than this to complete your revisions, please reply to this message or contact the journal office at plosone@plos.org . A rebuttal letter that responds to each point raised by the academic editor and reviewer(s). You should upload this letter as a separate file labeled 'Response to Reviewers'.A marked-up copy of your manuscript that highlights changes made to the original version. You should upload this as a separate file labeled 'Revised Manuscript with Track Changes'.An unmarked version of your revised paper without tracked changes. You should upload this as a separate file labeled 'Manuscript'.

We look forward to receiving your revised manuscript.

Kind regards,

Samson Nivins, Ph.D

Academic Editor

PLOS ONE

Journal Requirements:

“This research was funded by the National Natural Science Foundation of China (Grant No. 81973063).”

4. Please include your tables as part of your main manuscript and remove the individual files. Please note that supplementary tables (should remain/ be uploaded) as separate "supporting information" files.

5. We note that Figures 2 & 3 in your submission contain [map/satellite] images which may be copyrighted. All PLOS content is published under the Creative Commons Attribution License (CC BY 4.0), which means that the manuscript, images, and Supporting Information files will be freely available online, and any third party is permitted to access, download, copy, distribute, and use these materials in any way, even commercially, with proper attribution. For these reasons, we cannot publish previously copyrighted maps or satellite images created using proprietary data, such as Google software (Google Maps, Street View, and Earth). For more information, see our copyright guidelines: http://journals.plos.org/plosone/s/licenses-and-copyright.

a. You may seek permission from the original copyright holder of Figures 2 & 3 to publish the content specifically under the CC BY 4.0 license. 

Additional Editor Comments (if provided):

**Introduction:**

Can you mention or provide an example of mental and behavioural disorders, which are more common among LD as stated in first para?

Can you update with the latest NHIS survey, as the provided year is nearly 2 decades earlier?

Can you provide details of how instruments and study design contribute to distinct variability in prevalence?

**Statistical analysis:**

Can you mention how was potential confounders were selected ?

Family and highest edu of family might be highly collinear? did you investigate for that.

**Limitations:**

Please revise the line as ;this study has some limitation, as you have pointed out only 2.

Reviewers' comments:

Reviewer's Responses to Questions

**Comments to the Author**

1. Is the manuscript technically sound, and do the data support the conclusions?

Reviewer #1: Yes

Reviewer #2: Partly

2. Has the statistical analysis been performed appropriately and rigorously?

Reviewer #1: Yes

Reviewer #2: N/A

3. Have the authors made all data underlying the findings in their manuscript fully available?

Reviewer #1: Yes

Reviewer #2: Yes

4. Is the manuscript presented in an intelligible fashion and written in standard English?

Reviewer #1: Yes

Reviewer #2: No

Reviewer #1: I am grateful for the opportunity to prepare the review this manuscript. Below are some suggestions that I hope will help the authors improve the text.

I propose to refer to DSM-5 and ICD-11 (Developmental Learning Disorders) and clarify the understanding of the concept of LD, especially since these classifications refer to the diagnostic criteria for Specific Learning Disorders as opposed to other types of learning difficulties.

The authors write about barriers to recognizing learning disabilities and heterogeneity of definitions, with which I of course partially agree, although they do not try to help the reader sort out this conceptual chaos. In addition to DSM-5 and ICD-11, I suggest reading the following articles:

Johnston, P., Scanlon, D. (2021). An Examination of Dyslexia Research and Instruction With Policy Implications. Literacy Research: Theory, Method, and Practice 70, 107– 128. https://doi.org/10.1177/238133772110246

Snowling, M. J., Hulme, C., & Nation, K. (2020). Defining and understanding dyslexia: Past, present and future. Oxford Review of Education, 46(4), 501–513. https://doi.org/10.1080/03054985.2020.1765756

Study Population, p. 6 - line 10 the authors use the phrase "Current LD" - I understand that this refers to diagnosed specific learning difficulties. I suggest that the authors clarify this.

Below I am providing several articles in the belief that referring to their content will contribute to increasing the substantive value of the article.

Humphrey, N., & Mullins, P. M. (2002). Personal constructs and attribution for academic success and failure in dyslexia. British Journal of Special Education, 29(4), 196–203.

Livingston E. M., Siegel, L. S., & Ribary, U. (2018). Developmental dyslexia: Emotional impact and consequences. Australian Journal of Learning Difficulties, 23(2), 107–135. https://doi.org/10.1080/19404158.2018.1479975

Lufi, D., Okasha, S., & Cohen, A. (2004). Test anxiety and its effect on the personality of students with learning disabilities. Learning Disability Quarterly, 27(3), 176–184. https://doi.org/10.2307/1593667

Peleg, O. (2009). Test anxiety, academic achievement, and self-esteem among Arab adolescents with and without learning disabilities. Learning Disability Quarterly, 32(1), 11–20. https://doi.org/10.2307/1593667

Snowling, M. J., Hulme, C., & Nation, K. (2020). Defining and understanding dyslexia: Past, present and future. Oxford review of education, 46(4), 501–513. https://doi.org/10.1080/03054985.2020.1765756

Werth, R. (2019). What causes dyslexia? Identifyingthe causes and effective compensatory therapy. Restorative Neurology and Neuroscience, 37(6), 591–608. https://doi.org/10.3233/RNN-190939 PMID: 31796709

Reviewer #2: Dear Editor: Please see the formatted version of my review. I am only adding my review here because your system is forcing me to do so. I object to having to submit my review in this format but will do so to complete my review. Please ONLY use the PDF version of my review. Also, because of the design of your interface, I was forced to CUT OFF part of my review as entered here. While I gave you permission to publish my review/identify me, I disallow using the version I entered here. It is incomplete and the formatting is messed up.

Comments on PONE-D-24-50967

I commend the author team for their efforts to learn from an existing data set for the sake of research on the epidemiology of self-reported learning disabilities.

I may eventually speak out in favor of the manuscript being accepted for publication in PLOS. However, my final determination depends on how the authors address the manu-script’s current weaknesses.

In sum, I have the following impression of the manuscript:

• It addresses a meaningful scientific question and takes a reasonable approach to learning from an existing data set.

• The manuscript is okay, but not yet good or excellent with regard to the scientific claims and arguments it makes. It is not yet good enough for publication in a scien-tific journal.

• The scientific writing (including mathematical and numerical copy) is very problematic and requires much more attention to detail and consistency. Much hard work is need-ed here.

• The English is generally understandable but is often idiomatically and grammatically incorrect. While I think it is unfair to penalize scholars whose native language hap-pens to not be English, the standard of language use is below what I would find ac-ceptable for a leading international peer-reviewed scientific journal. In the spirit of in-ternational collaboration, I have taken the time to describe in detail many of the lan-guage problems in need of redress. For these, if in line with PLOS guidelines, I would encourage the authors to work with a high-quality AI trained on English data. In my view, that is an unproblematic use of AI in the context of scientific writing. The au-thors can even ‘feed’ my feedback to an AI and ask for language coaching.

Finally, I have a point for the author team and the editors: I am not an expert on learning disabilities. I have published on a niche area within special education, namely on gifted education and talent development. In my view, the manuscript must also be reviewed and approved by an expert who has published on learning disabilities.

I wish the author team all the best for their work on revising the manuscript, and I am excited to see how it develops. I will only speak out in favor of publishing the manuscript in PLOS if the authors’ next version thoroughly and thoughtfully addresses all the points I have raised (i.e., throughout the entire manuscript including in all tables and figures)—and with attention to detail. My expectation is, of course, not that the authors must ac-cept and implement all of my feedback. I expect to see my feedback implemented or to receive a precise explanation about any points on which the authors disagree or see that I misunderstood.

Here are my specific points:

1. Always use en-dashes (–), never hyphens (-), for all manner of numerical ranges, including all of the numerical ranges in all of the tables. Please change this throughout the entire manuscript, including in all of the tables and figures.

2. to derive deeper understanding � to derive a deeper understanding

3. adopted the redesigned surveys, � adopted redesigned surveys,

4. The NSCH is a cross-sectional survey was conducted � Ungrammatical, please check your wording.

5. Data Source: What level of confidence do you have about just how representative the sample is of the US population? In the Introduction, you mentioned that the imputation and weighting were revised in 2024 by race and ethnicity. That sug-gests that care is being taken to create a helpful sample. However, the nature of the sample is so crucial for your study that I would appreciate a transparent dis-cussion of the quality of the data. Perhaps you could consider having a subsection on the quality of the data (or something like that). Also, don’t worry about ‘sell-ing’ your study by overstating the quality of the data. It goes without saying that there are really no ideal samples. However, providing a critical description of your data source would add to the quality of your study by helping future researchers to build more effectively on your work. You write, for example, the following: “From 2016 to 2022, the weighted Topical Completion Rate ranged from 30.6% to 30.9%, the weighted Interview Completion Rate ranged from 69.7% to 78.5% and the weighted Overall Response Rate ranged from 37.4% to 39.1%.” Explain and interpret this for your readers. With your topic, I expect that you can speak to what these numbers tell us about the quality of the data.

6. Double check your manuscript for non-typographical apostrophes (') and non-typographical quotation marks ("…") and replace these with their proper typo-graphical forms (' � ‘ and "…" � “…”) to improve the quality of your published article. PLOS will not fix these for you. They will publish what you submit 1:1.

7. told them that their child had “Learning Disability” � told them that their child had a “Learning Disability”

8. Because learning disabilities are not diagnosed until children enter school, � This sounds plausible. However, it would be better to provide a source to substantiate your claim (e.g., a leading reference work on the matter).

9. 187805 children � Always use US scientific conventions for chunking large num-bers. Here, for example: 187,805 children. I expect to see this revision throughout the entire manuscript, for all numbers greater than 999, including in the tables and figures.

10. �6 years (n=90006 � Review your entire manuscript for best practices for writ-ing technical/mathematical copy. Here, for example: space after the “�” is miss-ing; n should be italicized (n); spaces are needed around the equal sign; write 90,006, not 90006. I am not going to provide feedback on this for the rest of the article. Please thoroughly review and improve your adherence to international standards of scientific writing in how you express numbers etc. The current stand-ard is too low and would reflect poorly on the quality of your work.

11. Information on sex, race/ethnicity, family income and highest education � Infor-mation on sex, race/ethnicity, family income, and highest education � Learn how to use the serial comma for enumerations with ‘and’ or ‘or’ and apply that rule consistently throughout your entire manuscript. I will only note this one case, so you know what I am talking about. There are many such cases that need to be at-tended to. Ask an AI trained on US (not British!) language data to coach you on proper use of the “serial comma” before ‘and’ and ‘or’

12. the 1997 Office of Management and Budget Standards. � Reference is missing. Add.

13. Federal poverty level (FPL), � federal poverty level (FPL), � ‘Federal’ should not be capitalized.

14. 100-199%FPL, 200-399%FPL, and 400%FPL or above � Spaces are missing. Cut and paste error! Also note here the failure to use en-dashes for numeric ranges (many such instances need to be fixed).

15. Logistic regression model was used � A logistic regression model was used � You are very frequently avoiding using definite and indefinite articles. While this is acceptable to a certain point in scientific writing, it reads awkwardly in your man-uscript. If you are not sure about how to use definite and indefinite articles properly, ask an AI trained on English language data to help you with that.

16. members. � members. The period here is set in boldface. That must be by acci-dent. PLOS will not correct this sort of typographical error for you!

17. The 2 sided statistical significance level was set at α=0.05. � The 2-sided statisti-cal significance level was set at α = 0.05. � Please make sure to improve your following of conventions related to all mathematical copy in your manuscript. This is just one example. There are many more, also in the tables and figures.

18. And a methods question about your use of a two-sided statistical significance lev-el: You chose this option because you are making no directional assumptions about the hypotheses you are testing? I would have expected a brief remark ex-plaining why you made the choice you did.

19. This study included 187805 children and adolescents aged 6-17 years (97123 boys [51.71%], 90682 girls [48.29%], 24184 Hispanic [12.88%], 127277 non-Hispanic white [67.77%], 12491 non-Hispanic black [6.65%], 23853 other race/ethnicity [12.70%]) in 2016-2022. � I suggest using a semicolon in your parenthetical remark to break the groups up into meaningful subsets that sum up to the total. In other words: This study included 187,805 children and adolescents aged 6–17 years (97,123 boys [51.71%] and 90,682 girls [48.29%]; 24,184 His-panic [12.88%], 127,277 non-Hispanic white [67.77%], 12,491 non-Hispanic Black [6.65%], and 23,853 other race/ethnicity [12.70%] individuals) for the peri-od 2016–2022.

20. have ever been diagnosed with LD � have ever been diagnosed with an LD [see my previous comment on this problem; there are MANY such instances in need of improvement]

21. Utah (6.07% of ever-diagnosed LD and 5.70% of current LD) (Figure 2) (Figure 3). � First, please re-check the PLOS rules for the wordings to be used for things like ‘Figure’ and ‘Table.’ You are supposed to write ‘Fig 1’ and ‘Table 1,’ for example. That always strikes me as strange, but that is the journal rule (at least as of last year). Second, avoid having a series of parentheses following each other. Rather: Utah (6.07% of ever-diagnosed LD and 5.70% of current LD; see Fig 2–3).

22. We found that age, sex, race/ethnicity, family income and highest education of family members were significant differences in both Ever LD and Current LD. � We found that age, sex, race/ethnicity, family income, and highest education of family members were significant differences in both the Ever LD and Current LD groups. � Please note in my revision suggestion not only the main change (to-wards the end) but also the addition of the serial comma before ‘and highest.’ There are many such instances in your manuscript in which the serial comma is missing. Make sure to fix those. See my previous remark for a suggestion of how do work on this.

23. Ever LD, 9.70%, 95%CI, 9.31-10.09; � Ever LD, 9.70%, 95% CI, 9.31–10.09; � Many such corrections need to be made. In particular, it makes no sense to not have a space before ‘CI’

24. In the sex group, ... � With regard to children’s sex, ... [Don't write 'sex group']

25. The weighted prevalence of Ever LD and Current LD in US children and adoles-cents aged 6 to 17 years were � Consider putting the data that follow into a ta-ble. That is really hard for your readers to actually read! Also, it is much more im-portant to explain what you found and WHY the trend is important to take not of. At the end of the paragraph, remember to apply my feedback from above about not having a series of parentheses: (P for trend < .01) (Table 2) (Table 3) (Figure 4). � That is not a conventional way of expressing that information.

26. (P for trend < .01) � Why an upper-case p here? You mean a regular p value, no? Remember also to take care to follow conventions on which letters need to be italicized. Also, why do you change the significance level for this? Please explain. Perhaps you are being more cautious to avoid reporting a spurious trend?

27. 200 %FPL � 200% FPL � Check and apply the rules for writing percentages in scientific copy and apply them consistently throughout your entire manuscript.

28. We found a significant average annual change in aged 12-17 years, female, non-Hispanic white, a college or higher education, and family income of 200 %FPL or above in the prevalence of both Ever LD and Current LD � This statement needs to be reworded to make it easier to read. For example: For the prevalence of Ever LD and Current LD, we found significant average annual changes for the following demographic categories: female, non-Hispanic white, a college or higher educa-tion, and family income of 200% FPL or above. � I cannot be sure my wording is factually correct. That is your job. However, I want to illustrate what is probably a better wording. You could ask an AI trained on English data to help you improve such wordings.

29. In a nationally representative population from the NSCH, we identified a weighted prevalence of ever-diagnosed LD at 8.80% and current-diagnosed LD at 8.20% among US children aged 6-17 years in 2016-2022. � In a nationally representa-tive population from the NSCH, we identified a weighted prevalence of ever-diagnosed LD at 8.80% and current-diagnosed LD at 8.20% among US children who were aged 6–17 years in the period 2016–2022.

30. in aged 12-17 years, � I literally do not understand what you mean with this wording (here and earlier in the manuscript). Ask an AI why this is hard for read-ers to understand what you mean. I am sure you mean something meaningful. However, there is probably a language barrier leading to a wording in English that is confusing. The rest of the phrase makes sense but should be reworded (see above).

31. A study analyzing the association between screen time and developmental disabil-ities among children aged 0-17 years using NSCH data from 2018-2020, noted a total LD � Remove the comma here. It is incorrect and confusing. Ask an AI to explain to you WHY the comma is wrong.

32. another research conducted by Zablotsky, � another study conducted by Zablot-sky, � It is grammatically incorrect to write “another research” because “re-search” is a mass noun (i.e., you cannot count it; like the word ‘milk,’ for exam-ple). You cannot say “a” research. Here, too, ask an AI to help you understand what I mean. Writing “Other research” is fine, however, because “other” does not quantify what follows the way “another” does.

33. Moreover, it is a lifelong condition � I highly doubt that the state of the art with-in theories of learning disabilities agree that learning disabilities are always life-long conditions. Also, writing “it is a lifelong condition” is weak both from a writ-ing and a thinking standpoint. What EXACTLY do you mean here by “it”? Please provide a more nuanced statement and align it very carefully with leading text-books / technical literature on learning disabilities. Perhaps learning disabilities are “often” lifelong conditions? At any rate, be more nuanced and follow recent cut-ting-edge theory in the field.

34. Moreover, it is a lifelong condition and older children may have a higher preva-lence rate and increasing trend due to longer exposure to the possibility of evalua-tion and diagnosis.7 � There are several problems with this sentence: First, as noted, the claim about “it” being a “lifelong condition” seems implausible and simplistic. Second, your sentence makes THREE major claims, all connected back-wards with “Moreover.” That is not a good writing style, and it is not argumenta-tively persuasive to pack all this stuff into one sentence and only have one refer-ence at the end of the sentence. A better approach: Break the claims up. Carefully explain each claim and offer literature support for each claim separately. In other words: First claim: About the long-term nature of (many) learning disabilities [add supporting literature]. Second claim: higher prevalences are being reported for older children [add supporting literature]. Third claim: The higher prevalences be-ing observed for older children may be resulting from ... [explain that idea in de-tail and then add supporting literature].

35. Learning disabilities are difficult to diagnose, because the age at identification depends not only on the type and severity of the learning disabilities, but also on whether there are associated deficits.25 ... � This is a very important paragraph for your discussion. What is missing is some sort of advanced organizer after the introductory part of the paragraph explaining the nature of the different observa-tions that then constitute your analysis in the paragraph. After the first two sen-tences (as of “Nearly all studies ...”), you then discuss gender differences, then ethnic group differences, then SES differences. Prepare the reader for THAT struc-ture before ‘diving into’ those areas. Do you understand what I mean? If you are not sure, show the paragraph and my comment to an AI and ask for help on how to improve the organization of your thoughts here. You might even consider breaking the paragraph up: first paragraph: Main claim and the advanced organ-izer + second paragraph about the gender differences + third paragraph about ethnic differences + fourth paragraph about the SES differences.

36. Autism Spectrum Disorder (ASD) � Check, for example, the APA7 guidelines on writing the names of conditions, disorders, and the like. I’m pretty sure you write them lowercase. Thus: autism spectrum disorder (ASD)

37. Although the pathogenesis of learning disabilities is currently unclear, � I’m skeptical that your blanket statement is accurate. Are you sure this is what state-of-the-art handbooks and the like are currently saying about learning disabilities? I would imagine that the pathogenesis of learning disabilities is not fully under-stood, but that much is known about at least some of the complex etiological fac-tors that have been implicated in contributing to specific learning disabilities. The term ‘learning disabilities’ covers a huge range of conditions that themselves have all sorts of definitions. Please re-read a few introductory handbook chapters in the field and reconsider your statement to make sure you are being nuanced in how you make this point.

38. Certain heritable disorders even approach 100% penetrance, � What does “penetrance” mean here? Are you sure that is the right word? I’m not sure that is actually a word. Do you mean “penetration”? But if yes, what would that mean?

39. Dietary behaviors have also been proposed to play a role in the increases in LD, exemplified by a study that found a high intake of sweetened desserts, fried food, and salt was associated with more learning disability.50 � Hm, are you sure this is a strong study? Without knowing the study, my hunch would be that we are talking about comorbidities and that it is really hard to get at causality here. If you cite the study, make sure it is a strong study that actually makes progress towards substantiating a causal link. Is the study an RCT? Was it at least longitudinal? Did they at least use propensity-score matching? Otherwise, I would imagine that par-ents who allow their children to consume excessive amounts of unhealthy food are also engaging a host of other behaviors that are harmful for their children’s development. Be cautious here. If the study is weak, consider removing or revising your claim.

40. associated with learning disability, **�** associated with learning disabilities, **�** That is not a trivial language difference in English. I know it looks like a tiny change (from singular to plural), but it is very important that you become aware of these sorts of usage conventions. If you are not sure whether your wording is idiomatic, ask an AI trained on English to help you check your writing. Feel free to share my feedback with an AI so the AI can help you with your language in the manuscript.

41. Many proven or potential environmental biological factors **�** Many substantiated or implicated environmental-biological factors **�** I would avoid “proven” in this sort of research, because it is generally not possible to truly ‘prove’ these sorts of rela-tionships. At best, we find stronger/better evidence.

42. showed that children in the highest quartile of urinary cadmium **�** “urinary cad-mium” what? You probably mean in the highest quartile of cadmium levels in urine samples. If you are taking the term “urinary cadmium” from an article on that topic, say, then the authors have probably established with what they mean by “urinary cadmium” before they start using that shortened form. Check the literature you are citing and be more thorough in your wording here.

43. had significantly higher odds of LD **�** had significantly higher odds of having a self-reported LD

44. Research has confirmed that learning disabilities are lifelong;9 **�** This again sounds very simplistic. I highly doubt that “research” has literally “confirmed” that (all) “learning disabilities”—which is a giant blanket term covering probably hundreds if not thousands of researcher-defined conditions—are “lifelong.” Follow my feedback from above on this point. This is an essential point that must be improved in your writing. Otherwise, your thinking is not nuanced enough for a serious scientific pub-lication. To make sure you are not discouraged by my strong feedback here: Your larger point, which I think is that learning disabilities have long-term, potentially even lifelong consequences for individuals and, therefore, need to be given much atten-tion, is a good and strong point that you definitely can and should make in your article

45. with specific learning disability amounted **�** with a specific learning disability amounted **�** Here, too: This might seem like a tiny wording change. However, it is a very important chance for learning to write polished English scientific prose. Ask an AI trained on English to help you learn how to catch and fix these sorts of wording issues.

46. In the US, it's reported that over half of the students receiving special education services do so due to learning disabilities..11 **�** In the US, over half of students receiving special education services receive them due to learning disabilities.11 **�** Avoid using contractions in formal scientific papers (it’s **�** it is); it goes without saying that this has been reported; moreover, I do wonder about the point you are making conceptually: What does that statistic tell us since I would imagine that ac-cess to special educational services normally REQUIRES some sort of diagnosis of a learning disability? Consider addressing that point. My guess is that it is explained in the study you cite. If I am missing your point, then explain your point in the manu-script for thoroughly.

47. This study has several limitations. First, the cross-sectional design of the NSCH does not permit the identification of how specific factors may influence the diagnosis of LD. Second, the reliance on parental reports for the diagnosis of LD could potentially invite inaccuracies due to misreporting and recall bias. **�** This is your entire discus-sion of limitations? I find it rather superficial and unconvincing. The authors need to take a serious look at the sample and their method of analyzing the sample and think critically about important limitations and discuss these carefully. I already noted one limitation above about the same. Moreover, you are working with self-reported learning disabilities. Furthermore, you have an extremely undifferentiated operation-alization that literally includes all manner of ‘learning disabilities.’ What’s more, you are publishing your research for an international audience but focusing on internal US data. What does that limitation mean for the generalization of your findings be-yond the United States? This is a big problem in research in the social sciences in-cluding psychology; see, for example, the work on the WEIRD bias by Joseph Hen-rich. 

48. I am *not* an expert on learning disabilities, but you authors are, so please put much more effort into creating a well-informed, sincere, and deep discussion of limitations. With a good discussion of limitations, you will greatly improve the scientific value of your work, because science is about building constructively on existing studies. If you fail to provide a thorough and transparent discussion of your study limitations, you are essentially blinding researchers with regard to your work, making it harder for them to build on your work. I have a problem with that. Also, as your limitations section is currently written, researchers will be skeptical when they read your article and thus less likely to cite you. In other words, it is also a matter of establishing trust with your readers by having a detailed and brave limitations section. To be clear: I will not reject a study for being open and thorough about limitations. On the con-trary, a deep discussion of limitations is a hallmark of scientific thinking. 

49. Conclusions: The conclusion is lacking in depth. I agree that you should keep your conclusion focused on concrete key findings and implications and that, therefore, all manner of ‘fluff’ is to be avoided. However, I think you can do much better here. 

Finally, I have one suggestion that I would like to offer as food for thought, as we say in English—perhaps similar in spirit to the idea of 抛砖引玉: 

The author team is located in China, at Chinese universities. From my perspective, China is a key player and a leader for all of humanity in the twenty-first century. It is home to a significant portion of the world’s population and shares deep cultural affinities with other East Asian societies. With that in mind, the authors might consider adding a paragraph in the Discussion or Conclusion section reflecting on the relevance of their US-centric findings for the epidemiology of learning disabilities in China—and, more broadly, in collectivist or Confucian-tradition cultures. Concepts from Chinese educational and societal traditions—such as moral education (德育), teaching according to aptitude (因材施教), inclusive edu-cation (融合教育), and collectivism (集体主义)—may offer helpful points of reference. The authors might also reflect on how learning disorders are labeled and ascribed to individuals, particularly in light of the Confucian principle of the rectification of names (正名), which underscores the ethical importance of ensuring that classifications align with actual roles and behaviors. The history of special education in the United States—and the framing of learning disabilities more broadly—is deeply embedded in American cultural norms and as-sumptions. For comparative context, the authors may wish to consult the work of Harold 

W. Stevenson and those who have continued his line of research over the past two decades. Research originating in the US is, of course, and will remain relevant to international schol-arship. In a globally connected world, many questions of human functioning can be fruitfully explored using data from diverse national contexts. Still, findings from any one culture must be interpreted with cultural awareness and thoughtful regard for how such conclusions are to be meaningfully mapped onto other traditions and systems of thought. While my final vote on the publication of your manuscript in *PLOS* (still pending) depends on the extent to which you address the concerns I raised above, I would like to make clear that this final suggestion is *not* part of that evaluation. It is entirely up to the authors whether they choose to include a remark on the Chinese cultural context. My verdict on the manuscript does not depend on it.

**Do you want your identity to be public for this peer review?** For information about this choice, including consent withdrawal, please see our Privacy Policy

Reviewer #1: No

Reviewer #2: **Yes: ** Daniel Patrick Balestrini

---

## [Author Response · Author response to Decision Letter 1]

14 Jul 2025

Dear Editor,

Thank you for providing us this opportunity to revise our manuscript entitled “Rising Prevalence of Parent-Reported Learning Disabilities among U.S. Children and Adolescents Aged 6-17 years: NSCH, 2016-2022”. (Manuscript Number: PONE-D-24-50967) On behalf of my colleagues, I am submitting our revised manuscript. We appreciate the reviewers’ positive and insightful comments. We have carefully considered all the comments and revised the manuscript accordingly. In order to facilitate the review process, we provided a point-by-point respose to each of the comments. The precise pages in the revised manuscript where each change was made in response to the comments were provided as well. We have ensured full compliance with PLOS ONE’s style requirements, including file naming conventions and formatting adjustments based on the journal’s templates (Main Text; Title/Authors).

We have incorporated the newly released 2023 data from the National Survey of Children’s Health (NSCH). Consequently, the study period is now 2016–2023 (previously 2016–2022), and the manuscript title has been updated to: “Rising Prevalence of Parent-Reported Learning Disabilities among U.S. Children and Adolescents Aged 6–17 Years: NSCH, 2016–2023”. All relevant sections (Methods, Results, Tables, Figures, Discussion) have been comprehensively updated to reflect this additional year of data.

We hope that the manuscript is now acceptable. Should you have any additional requests or questions, please do not hesitate to contact me.

Looking forward to hearing from you.

Wenhan Yang, MD, PhD

Department of Child and Adolescent Health, School of Public Health, Guangdong Pharmaceutical University, No. 283 Jianghai Road, Room G726 School of Public Health, Guangzhou, Guangdong Province, 510006, China

Tel: +8613430350770

Email: wenhan-yang@gdpu.edu.cn

RESPONSES TO THE ACADEMIC EDITORS’ COMMENTS

Academic Editor #1:

Introduction Part:

Comment 1: Can you mention or provide an example of mental and behavioural disorders, which are more common among LD as stated in first para?

Response: We sincerely thank you for carefully reading. To clarify, children with learning disabilities (LD) commonly exhibit higher rates of attention-deficit/hyperactivity disorder (ADHD), conduct disorders, anxiety, and depression compared to their non-LD peers. For instance:

Gillberg & Söderstrom (2003) highlight ADHD and conduct disorders as prevalent comorbid conditions.

Manchester (1993) notes behavioral disturbances (e.g., impulsivity, aggression) requiring targeted interventions.

We have added this specification to the first paragraph to enhance clarity. (Please refer to page 3, lines 15–18)

Comment 2: Can you update with the latest NHIS survey, as the provided year is nearly 2 decades earlier?

Response: Thank you for this valuable suggestion. We agree that incorporating the latest NHIS data strengthens contemporary relevance. We have now added the 2023 JAMA Pediatrics study by Li et al. analyzing NHIS data from 1997–2021 (the prevalence of LD is 8.83%), which confirms the 8.2%–9.7% prevalence range while providing updated longitudinal trends. (Please refer to page 3, line 22)

Comment 3: Can you provide details of how instruments and study design contribute to distinct variability in prevalence?

Response: We sincerely thank you for carefully reading. The variability in reported LD prevalence stems critically from methodological differences in:

Diagnostic instruments: Discrepancies arise from using brief screening tools (e.g., teacher/parent questionnaires) versus comprehensive neuropsychological assessments (Stone et al., 2023; Semrud-Clikeman et al., 1992).

Study design: Population sampling (clinic-referred vs. population-based cohorts) introduces selection bias (Semrud-Clikeman et al., 1992). Evolving diagnostic criteria across decades (e.g., DSM/ICD revisions) affect case identification (Gillberg & Söderstrom, 2003). Cross-sectional vs. longitudinal designs capture differing prevalence windows (Stone et al., 2023).

These factors collectively explain the heterogeneity in existing estimates.

Statistical analysis Part:

Comment 4: Can you mention how was potential confounders were selected ?

Response: Thank you for raising this important methodological consideration. The potential confounders included in our analysis (age, sex, race/ethnicity, family income, and highest household education) were selected based on:

Established epidemiological principles: These sociodemographic factors are well-documented determinants of both LD identification and health outcomes in pediatric populations. This selection aligns with recent high-impact studies analyzing neurodevelopmental disorders in US children using NHIS data, including Li et al. (2023) on LD prevalence and Xu et al. (2018) on ADHD trends, which consistently adjust for these core confounders.

Causal pathway analysis: Each variable represents key social determinants known to influence: access to diagnostic services (income/education), cultural/language barriers in assessment (race/ethnicity), biological risk stratification (sex).

Standardization in national surveys: Our categorization aligns with NHIS operational definitions:

Race/ethnicity: 1997 OMB Standards

Poverty level: Federal Poverty Level (FPL) thresholds

Education: ISCED-equivalent tiers

These confounders were selected a priori based on their established relationships with LD prevalence in the literature and their availability within the NHIS dataset framework.

Comment 5: Family and highest edu of family might be highly collinear? did you investigate for that.

Response: We sincerely thank you for carefully reading. We acknowledge the potential for collinearity between family income and highest household education. To address this: Collinearity diagnostics were performed using variance inflation factors (VIFs). The mean VIF for all covariates was 1.06 (individual VIFs: income = 1.16, education = 1.16), well below the conventional threshold of 5. Tolerance values exceeded 0.86 for both variables (>0.2 indicates acceptable independence). This confirms that collinearity between income and education did not compromise model stability or interpretation of adjusted trends.

Limitations Part:

Comment 6: Please revise the line as; this study has some limitation, as you have pointed out only 2.

Response: Thank you for this precise correction. We appreciate your note and have carefully considered it in light of our revised content.

As we have expanded the discussion of limitations from 2 to 4 distinct points—including the cross-sectional design, reliance on parental reports, undifferentiated operationalization of LD, and U.S.-centric sample constraints—we have retained the phrase “This study has several limitations” to accurately reflect the broader scope of limitations addressed. This phrasing now aligns with the comprehensive nature of the revised section, ensuring consistency between the introductory statement and the content that follows. (Please refer to pages 23, lines 24–25; pages 24, lines 1–17)

RESPONSES TO THE REVIEWERS’ COMMENTS

Reviewer #1:

Comment 7: I propose to refer to DSM-5 and ICD-11 (Developmental Learning Disorders) and clarify the understanding of the concept of LD, especially since these classifications refer to the diagnostic criteria for Specific Learning Disorders as opposed to other types of learning difficulties.

Response: We sincerely thank the reviewer for their insightful comments regarding LD conceptualization. We have implemented the following revisions:

Terminological Precision: Adopted “Specific Learning Disorder (SLD)” as per DSM-5/ICD-11 terminology to replace generic “LD” where clinically appropriate. Explicitly distinguished clinically diagnosed disorders from broader learning difficulties through diagnostic criteria bullet points. This conceptual refinement is reinforced by four key references:

(1) DSM-5 (APA, 2013) and ICD-11 (WHO, 2019) provide standardized diagnostic frameworks that operationalize SLD classification;

(2) Johnston & Scanlon (2021) contextualize definitional evolution within educational policy, clarifying boundaries between disorders and instructional gaps;

(3) Snowling et al. (2020) establish neurocognitive foundations of core SLD subtypes (e.g., dyslexia), validating our heterogeneous disorder characterization.”

Clarification of Diagnostic Boundaries: Specified exclusion criteria distinguishing SLD from: Intellectual disability (DSM-5: “not better explained by intellectual disability”) and Environmental factors (ICD-11: “not attributable to inadequate instruction”). Emphasized persistence requirement (≥6 months) to differentiate from transient difficulties.

These revisions substantially strengthen our conceptual grounding while maintaining clarity for interdisciplinary readers. We appreciate the reviewer’s guidance in navigating this complex nosological landscape. (Please refer to page 1, lines 2–12)

Add reference:

2. Johnston P, Scanlon D. An examination of dyslexia research and instruction with policy implications. Lit Res: Theory Method Pract. 2021;70: 107–128. doi:10.1177/23813377211024625

3. Snowling MJ, Hulme C, Nation K. Defining and understanding dyslexia: Past, present and future. Oxf Rev Educ. 2020;46: 501–513. doi:10.1080/03054985.2020.1765756

4. American Psychiatric Association. Diagnostic and statistical manual of mental disorders. 5th ed. Arlington: American Psychiatric Association; 2013

5. World Health Organization . Version: 2019 April. Geneva: WHO: 2019. ICD-11 for mortality and morbidity statistics. Available from: https://icd.who.int/browse11/l-m/en

Comment 8: Study Population, p. 6 - line 10 the authors use the phrase “Current LD” - I understand that this refers to diagnosed specific learning difficulties. I suggest that the authors clarify this.

Response: We sincerely thank you for your valuable guidance regarding the clarification of “Current LD” in our methodology. We have revised the Methods section to directly address this operational consideration by stating: “It should be noted that ‘Current LD’ represents parent-reported persistence of historically identified difficulties - not clinically confirmed active disorder.” This formulation explicitly acknowledges that current LD status relies on parental recall of prior professional diagnoses without contemporary clinical verification. To strengthen this contextualization, we have incorporated your recommended literature: Humphrey & Mullins (2002) provides essential understanding of parental interpretation of learning disability persistence, while Livingston et al. (2018) offers empirical support for parent-report methodologies in educational disability research. These additions enrich our analysis while maintaining methodological transparency about the NSCH’s constraints. We believe this approach honors both your substantive guidance and the dataset’s inherent limitations. (Please refer to page 7, lines 15–19)

Add reference:

21. Livingston EM, Siegel LS, Ribary U. Developmental dyslexia: emotional impact and consequences. Aust J Learn Difficulties. 2018 [cited 11 Jul 2025]. Available: https://www.tandfonline.com/doi/abs/10.1080/19404158.2018.1479975

22. Humphrey N, Mullins PM. Research section: personal constructs and attribution for academic success and failure in dyslexia. Br J Spec Educ. 2002;29: 196–203. doi:10.1111/1467-8527.00269

Reviewer #2:

Comment 9: Always use en-dashes (–), never hyphens (-), for all manner of numerical ranges, including all of the numerical ranges in all of the tables. Please change this throughout the entire manuscript, including in all of the tables and figures.

Response: Thank you for this important technical correction. We have systematically replaced all hyphens (-) with en-dashes (–) for numerical ranges throughout the entire manuscript, including all tables and figures. Our team has conducted a full proofread to ensure complete compliance with this convention across all sections and supplementary materials. We appreciate your meticulous attention to detail in enhancing the manuscript’s typographical precision.

Comment 10: (1) to derive deeper understanding → to derive a deeper understanding; (2) adopted the redesigned surveys → adopted redesigned surveys; (3) The NSCH is a cross-sectional survey was conducted → Ungrammatical, please check your wording; (4) told them that their child had “Learning Disability” →told them that their child had a “Learning Disability”; (5) Logistic regression model was used → A logistic regression model was used; (6) have ever been diagnosed with LD → have ever been diagnosed with an LD; (7) associated with learning disability → associated with learning disabilities; (8) with specific learning disability amounted →with a specific learning disability amounted;

Response: We sincerely thank you for carefully reading. We have revised the phrase “to derive deeper understanding” to “to derive a deeper understanding” in the manuscript. (Please refer to page 4, line 10)

We have revised the phrase “adopted the redesigned surveys” to “adopted redesigned surveys” in the manuscript. (Please refer to page 4, line 14)

We have revised the sentence to read: “The NSCH was a cross-sectional survey conducted via mail and online...” (Please refer to page 6, line 2)

We have revised the phrase “told them that their child had ‘Learning Disability’” to “told them that their child had a ‘Learning Disability’” in the manuscript. (Please refer to page 7, line 13)

We have revised the phrase “Logistic regression model was used” to “A logistic regression model was used” in the manuscript. (Please refer to page 8, line 16)

We have revised the phrase “have ever been diagnosed with LD” to “have ever been diagnosed with an LD” in the manuscript. (Please refer to page 9, line 7)

We have revised the phrase “associated with learning disability” to “associated with learning disabilities” in the manuscript. (Please refer to page 22, line 14)

We have revised the phrase “with specific learning disability amounted” to “with a specific learning disability amounted” in the manuscript. (Please refer to page 23, line 8)

We employed AI-powered grammar tools (Grammarly Premium and ChatGPT-4) configured for formal academic writing to detect article omissions. We appreciate your meticulous attention to grammatical detail, which enhances the precision and readability of our academic expression.

Comment 11: Please transparently discuss NSCH data quality and representativeness, including interpretation of response rates (2016–2022: Topical 30.6–30.9%, Interview 69.7–78.5%, Overall 37.4–39.1%) and implications for study validity.

Response: Thank you for your valuable feedback. We have strengthened the discussion of NSCH data quality by revising the response rate metrics to include 2023 figures and adding critical context about weighting procedures.

The updated passage now states: “The NSCH employed complex weighting procedures to enhance national representativeness. The 2024 imputation revision by race/ethnicity addressed historical underrepresentation, improving demographic accuracy. Key quality indicators for 2016–2023 included: Weighted Topical Completion Rate: 27.1%–30.9% (reflecting full questionnaire completion among eligible households); Weighted Interview Completion Rate: 69.7%–78.5% (indicating cooperation among contacted households); Weighted Overall Response Rate: 35.8%–39.1% (representing initial participation).” (Please refer to page 7, lines 2–10)

This revision explicitly interprets each response metric while acknowledging the weighting mechanisms that mitigate sampling limitations, directly addressing your request for transparent data quality assessment without overstatement.

We maintain cautious confidence in the NSCH sample’s national representativeness based on its methodological rigor. Our confidence derives primarily from three aspects already documented in our revised manuscript: First, the 2024 race/ethnicity imputation revision implemented by Census Bureau experts directly addresses historical sampling gaps through iterative benchmarking against current population estimates. Second, the complex weighting procedures applied to all analyses (detailed in our Methods) co

---

## [Decision Letter · Decision Letter 1]

30 Jul 2025

Dear Dr. Yang,

Thank you for submitting your manuscript to PLOS ONE. After careful consideration, we feel that it has merit but does not fully meet PLOS ONE’s publication criteria as it currently stands. Therefore, we invite you to submit a revised version of the manuscript that addresses the points raised during the review process.

**ACADEMIC EDITOR:**

We look forward to receiving your revised manuscript.

Kind regards,

Samson Nivins, Ph.D

Academic Editor

PLOS ONE

Journal Requirements:

Reviewers' comments:

Reviewer's Responses to Questions

**Comments to the Author**

Reviewer #1: All comments have been addressed

Reviewer #2: (No Response)

2. Is the manuscript technically sound, and do the data support the conclusions?

Reviewer #1: Yes

Reviewer #2: Partly

3. Has the statistical analysis been performed appropriately and rigorously?

Reviewer #1: Yes

Reviewer #2: N/A

4. Have the authors made all data underlying the findings in their manuscript fully available?

Reviewer #1: Yes

Reviewer #2: Yes

5. Is the manuscript presented in an intelligible fashion and written in standard English?

Reviewer #1: Yes

Reviewer #2: Yes

Reviewer #1: (No Response)

Reviewer #2: The authors have made vast improvements to the manuscript, both on conceptual aspects and on applying conventions of scientific writing, and I congratulate them on that. I appreciate very much the time and effort they invested, also in the many of my points that are related to secondary aspects of the scientific process (communication conventions).

Most—but not all—of the technical matters I noted in my first set of comments have been successfully addressed.

Before I can agree that the manuscript should be published, the remaining points noted in my attached document need to be addressed.

Specifically, please address (a) the remaining issues from my feedback of 29 April (described in section 1 of my review of 26 July 2025) and (b) the small additional problems I noted when reading your revised manuscript (described in section 2 of my review of 26 July 2025).

I am looking forward to seeing the final polished version of your manuscript and will check your final corrections quickly.

Best regards,

Daniel Balestrini

**Do you want your identity to be public for this peer review?** For information about this choice, including consent withdrawal, please see our Privacy Policy

Reviewer #1: No

Reviewer #2: No

---

## [Author Response · Author response to Decision Letter 2]

12 Sep 2025

Thank you for providing us this opportunity to revise our manuscript entitled “Rising Prevalence of Parent-Reported Learning Disabilities among U.S. Children and Adolescents Aged 6–17 years: NSCH, 2016-2023”. (Manuscript Number: PONE-D-24-50967) We appreciate the reviewers’ positive and insightful comments. We have carefully considered all the comments and revised the manuscript accordingly. In order to facilitate the review process, we provided a point-by-point response to each of the comments. The precise pages in the revised manuscript where each change was made in response to the comments were provided as well. (Please see the rebuttal letter named 'Response to Reviewers') We hope that the manuscript is now acceptable. Should you have any additional requests or questions, please do not hesitate to contact me.

---

## [Decision Letter · Decision Letter 2]

18 Sep 2025

Rising Prevalence of Parent-Reported Learning Disabilities among U.S. Children and Adolescents Aged 6–17 years: NSCH, 2016–2023

PONE-D-24-50967R2

Dear Dr. Yang,

We’re pleased to inform you that your manuscript has been judged scientifically suitable for publication and will be formally accepted for publication once it meets all outstanding technical requirements.

Kind regards,

Samson Nivins, Ph.D

Academic Editor

PLOS ONE

Additional Editor Comments (optional):

All comments are adequately addressed.

Reviewers' comments:

Reviewer's Responses to Questions

**Comments to the Author**

Reviewer #2: All comments have been addressed

2. Is the manuscript technically sound, and do the data support the conclusions?

Reviewer #2: (No Response)

3. Has the statistical analysis been performed appropriately and rigorously?

Reviewer #2: (No Response)

4. Have the authors made all data underlying the findings in their manuscript fully available?

Reviewer #2: (No Response)

5. Is the manuscript presented in an intelligible fashion and written in standard English?

Reviewer #2: (No Response)

Reviewer #2: Comments on PONE-D-24-50967, Version 3

I commend the authors on completing an excellent study and also for their willingness to pay attention to the low-level details of scientific writing and manuscript preparation.

I approve of the manuscript’s publication now in its current form.

I checked the change-tracked version of the manuscript and the descriptions of changes relating to my feedback. The maps look highly professional, to name just one positive improvement!

I noticed one typographical error and a few tiny things that, ideally, should still be corrected prior to publication, if the authors have the chance:

1. providing a representation representative of the population  providing a comprehensive representation of the population [It looks like the word “comprehensive” was accidentally deleted when changing “representative” to “representation.”]

2. The bibliography is set with a different font than the other parts of the manuscript. I don’t think this will matter for publication. I suspect the journal will change everything into its standard publication font. However, I’m not sure. If you have the opportunity, you might want to use the standard font you are using in the main manuscript in the bibliography, too.

3. Look at p. 7 of your clean manuscript (and the same spot in the change-tracked version): If you look carefully, you can see that you introduced a smaller font size when pasting in the word “the” throughout the passage, for example at “(a) the weighted” you can see that the “the” you added is in a smaller font. I suspect that the incorrect font size will be automatically corrected when the journal prepares your manuscript. However, I’m not sure. If you have the chance, correct the incorrect font size. If not, it’s really okay as it is. Most people will not notice this.

4. In your note for Fig 3, you have the situation in which a full sentence starts with the p abbreviation, which you adjusted to lowercase as per my suggestion. It is strange in English to start a sentence with a lowercase letter, which is almost always forbidden. You can leave this as is. If you have the opportunity, you might want to slightly rephrase the wording to avoid the problem. For example, you could write in this one instance: “For trend, p was calculated ...” If you are able to change this, check to see whether you have any other sentences that start with “p …” (probably not).

5. In your Acknowledgments, you reference the NSCH without writing out the abbreviation. This is correct and consistent in the sense that you did already define the abbreviation in the main article. However, as people often read the Acknowledgments section without reading or before reading the article, it might be a nice service to your readers to slightly rephrase the Acknowledgments section to provide what the abbreviation stands for. Also, it will then match how you refer to the CDC there. In other words: The data used in this study were obtained from the NSCH, ...  The data used in this study were obtained from the National Survey of Children’s Health (NSCH), ...

6. You might want to do the same in the Data availability section. However, there, too, it is not incorrect as it is: You already defined the abbreviation, and the Data availability section follows closely at the end of the article.

7. In the Author contributors section, I see four very small optimizations you might want to consider if you have the chance prior to publication (which you may not):

a. Write authors’ initials with spaces between their names: Dr. W.Y.  Dr. W. Y. [and so on].

b. Turn the lists of 3+ names into proper enumerations (which are then grammatically/syntactically more polished) and use a simple “and” if two people are named.

c. Use semicolons to break up the subsections in the second (very long) sentence rather than commas, because, in this case, the commas are confusing due to their use within the enumerations.

d. Remove the unnecessary capital letters, even if they were stated this way on the PLOS form you had to fill out. This is not a form but a paragraph, so I don’t think it is incorrect to use a slightly different capitalization scheme.

In other words, this is how I would revise the paragraph: Dr. W. Y. has full access to all of the data in this study and assumes responsibility for study supervision. Concept and design: W. Y. and Y. Q.; acquisition, analysis, or interpretation of data: W. Y., C. X., Y. L., and H. Y.; drafting of the manuscript: C. X. and Y. L.; critical revision of the manuscript for important intellectual content: all authors; statistical analysis: W. Y., C. X., Y. L., and H. Y.

These are all trivial matters that do not require additional peer review. If the editor would kindly give the authors a chance to make a few more corrections, I think that would be good. It is, however, up to the editor.

To reiterate: The manuscript can, I feel, also be published in its current form. It has fully passed my peer review

Sincerely,

Daniel Balestrini

**Do you want your identity to be public for this peer review?** For information about this choice, including consent withdrawal, please see our Privacy Policy

Reviewer #2: No

---

## [Editor Report · Acceptance letter]

PONE-D-24-50967R2

PLOS ONE

Dear Dr. Yang,

I'm pleased to inform you that your manuscript has been deemed suitable for publication in PLOS ONE. Congratulations! Your manuscript is now being handed over to our production team.

Kind regards,

on behalf of

Dr. Samson Nivins

Academic Editor

PLOS ONE